# Differences between Creativity and Gender in Students with High Abilities Attending a School with Total Grouping

**DOI:** 10.3390/children9071081

**Published:** 2022-07-20

**Authors:** Julián Betancourt, María de los Dolores Valadez, Elena Rodríguez-Naveiras, Juan Francisco Flores, África Borges

**Affiliations:** 1Jalisco Educational Center for High Abilities, Jalisco Education Secretary, Guadalajara 44100, Mexico; coordinacion@cepac.edu.jalisco.gob.mx; 2Institute of Psychology and Special Education, Department of Applied Psychology, University Center of Health Sciences, University of Guadalajara, Guadalajara 44340, Mexico; dolores.valadez@academicos.udg.mx (M.D.V.); francisco.flores@academico.udg.mx (J.F.F.); 3Department of Clinical Psychology, Psychobiology and Methodology, University of La Laguna, 38200 San Cristóbal de La Laguna, Spain; aborges@ull.edu.es

**Keywords:** high ability, creativity, training

## Abstract

Background: Creativity is one of the most relevant aspects in students’ training. One of the purposes of the present work is to show the lack of differences between boys and girls in creativity; the other is the possibility of improving creativity among high-ability students who received specific training as part of their intra-curricular content in a total grouping program for gifted students. Method: The sample consisted of 42 students from first to third grade (13 females and 29 males) and 58 students from fourth to sixth grade (21 females and 37 males). Creativity was measured with the CREA test for younger students and with the PIC-N for older students. Training was carried out through an Integral Innovation and Creativity Program (PIIC). Results: The results showed no differences between genders, except in one of the graphic creativity scales (Details). There were improvements due to the training in all measures of narrative creativity and in the scale of elaboration of graphic creativity. Conclusions: The main conclusions are the importance of increasing creativity with specific programs and the need to improve interventions in graphic creativity.

## 1. Introduction

High-ability students clearly differ from their normative peers in terms of cognitive functioning. They have high curiosity, great memory [1], and high levels of abstract thinking, adaptation to new situations, and cognitive flexibility [2], as well as creativity [3,4]. However, their most distinctive characteristics are the large amount of information they handle, along with the greater speed of processing it and the use of metacognitive skills [5]. This leads them to prefer complex and challenging tasks, as well as to perform in elaborate learning environments where they deepen their knowledge and they address the diversified interests that they present [1,6].

According to the differentiation model, curricula and instruction should be adapted to students’ needs. Along this line, [7,8,9] mentioned that the raison d’être of educating high-ability students should be a differentiated curriculum. This paradigm strives for an optimal match between the potential and interests of the high-ability students and what should be offered to them. The Differentiated Model of Giftedness and Talent is an example of this conceptualization [10].

Therefore, these students require a specific educational response, which can be achieved through enrichment, acceleration, or grouping [11,12,13].

Grouping is a topic that has caused much disagreement. On the one hand, it has been considered to cause negative effects on students with high abilities, and, on the other hand, the benefits of this type of intervention have been mentioned to underline the greater evidence of its positive effects [14,15,16,17,18,19].

Grouping can occur in a variety of ways, such as within or between classes, for the whole school day, or for a reduced time, and it is often arranged in a flexible way according to students’ performance and their achievements [20,21,22].

Among the different types of grouping is cluster grouping (also known as total school grouping), where high-ability students are placed in a specific classroom with a teacher who is trained in high-ability and differentiated education [20,22,23]. This modality has been proven to have positive academic, social, emotional, and creative impacts. Regarding creativity, [18] mentioned that high-ability students benefit more from grouping, as it provides them with the opportunity to access deeper and more advanced knowledge, as well as to exercise creative and reflective thinking.

Regarding the latter, [18,24] stated that high-ability students benefit more from grouping, as it provides them with the opportunity to access deeper and more advanced knowledge and to exercise creative and reflective thinking. Likewise, [25] pointed out the importance of fostering creativity in these students in the school environment because it strengthens their ability to solve problems.

In Mexico, the Ministry of Education of the State of Jalisco opened the Educational Center for High Abilities (CEPAC) at the elementary level, which follows the modality of total grouping by abilities through innovative teaching methodologies associated with new information technologies and the use of a differentiated curriculum.

Creativity is indisputably important for innovation in any field and for facing everyday challenges [26,27]. Therefore, it is necessary to promote it from the early years of education—both at home and at school—to prepare students for the unknown, so that they can adapt to new situations and solve problems [28,29]. This implies that teachers should both educate creatively and foster creativity among their students [30]. According to [31], divergent thinking could be defined as a different way of processing information in which new connections are established that enable the creative process to solve problems in a different way from that in which most people do. It is mainly defined according to four skills: fluency, flexibility, originality, and elaboration.

Creativity can be defined as something new and original, and high quality and surprise are some of its components [32]. Although creativity has an indisputable value in the development and education of people [33], and despite the fact that the formal objective of the educational system is to facilitate the development of creative abilities and comprehensive care for all students, including the most gifted, the truth is that it is not promoted in either general education or for students with high abilities [34,35,36].

Implementing creativity in the curriculum promotes diverse ways of problem solving in students—in both their education and their daily life—producing a positive impact impact on their academic performance [37]. Several studies have found a positive relationship between these two variables [38,39,40,41,42]. However, it has not yet received the attention it deserves within the school setting. Creative thinking is necessary in order to promote a change in the different areas of science, technology, engineering, art, and mathematic (STEAM), which are all linked to the objectives of 21st-century education. In turn, working in the different areas of STEAM allows the development of critical and creative thinking [43,44,45].

Although it is assumed that, with respect to intelligence, there are no gender differences, there is still debate within the field of creativity on the existence of gender differences in this variable. [46] mentioned that, while social intelligence is dominant in women, creative intelligence is dominant in men.

There is a dispute within the field of creativity on the possible existence of gender differences. In a study where the nonverbal creativity was compared between adolescents from Mexico and Lithuania, the authors found that, in the visual–spatial category, females scored significantly higher than males in fluency, flexibility, and elaboration, whilst in the inventive category, males scored significantly higher in fluency and originality. [47] also found that girls obtained higher scores in creativity.

In their study, [48] reported that gifted male students excel in creativity, particularly in originality. In a recent study, it was found that women are more creative than men, particularly in flexibility; in the rest of the components of creativity, no significant differences were observed [49]. For their part, [50,51,52] found no differences according to gender. With these differences, it is not possible to have a clear view regarding gender and creativity. It is also important to highlight the different instruments used to measure general creativity, as well as the different types of creativity and its components.

It is unquestionable that the incorporation of activities and programs that stimulate creativity allows to increase it [53]. This result can also be obtained by introducing technology and robotics [54].

The aim of this research is to verify if there are any differences in creativity between boys and girls with high abilities and to check if specific training produces an increase in creativity.

## 2. Method

### 2.1. Participants

The participants in this study were students from first to sixth grade at the Educational Center for High Abilities (CEPAC (its Spanish acronym)). To join the Center, students underwent a selection process that included measures of intelligence, creativity, and personal and social adaptation (Betancourt, in preparation). Table 1 shows the elementary school members who entered CEPAC in the 2017 school year, at the time this study.

### 2.2. Instruments

The following instruments were used to measure creativity.

Creative Imagination Test for Children (CREA [55]). This provides an indirect measure of creativity. It covers ages from 6 years to adulthood. It consists of three stimulus sheets (A, B, and C); C was the one used in this study, since it is for the school population. From the stimulus sheet, the interviewees must formulate as many questions as possible in a period of 4 min. The CREA is an instrument that can be applied individually or collectively. The raw score obtained is transformed into a centile score while considering the relevant scales for each group and context. A reliability of 0.87 and a concurrent validity of 0.79 to 0.81 are reported. This test was used to collect data from students from first to third grade.Creative Imagination Test for Children (PIC-N, [56]). The test provides a total score for creativity and a score for narrative and graphic creativity. It also provides a score for each variable that makes up narrative creativity (fluidity, flexibility, and originality) and graphic creativity (elaboration, shadows and color, title, and special details). The PIC-N was aimed at children 8–12 years old; it had a reliability of 0.83. The correlation with a g-factor test was assessed, and correlations of 0.31 to 0.40 were reported. It was used to measure creativity in children from fourth to sixth grade.

The reliability values obtained in each scale for the studied sample are presented in Table 2.

### 2.3. Procedure

Authorization to collect data for the present study was obtained from the parents. In addition, authorization to conduct the study was requested from the Ethics Committee for Research and Animal Welfare of the University of La Laguna, and it was granted (CEIBA2020-0385).

To assess changes in creativity, tests were administered at the beginning and the end of the school year. During the school year, students took part in an Integral Innovation and Creativity Program (PIIC), which was based on the three types of enrichment established by Renzulli: awareness, concrete methodologies, and research and communication.

This program had an innovative and participatory methodology, which sought to generate other types of skills in the students; for this purpose, it moved away from a conventional and theoretical class and developed as a workshop. It was supported by audio–visual and technological resources, contacts, and networks in order for them to promote their own innovative and creative projects. Its aim was to develop creative and innovative thinking in students through the identification and resolution of problems that started from their real context.

Students attended the Creativity and Innovation Lab once a week for one hour to participate in the PIIC. It was implemented by a facilitator who was required to play an active role in its development and create a climate of trust in the classroom in order to foster new ideas.

### 2.4. Data Analysis

To test the differences in creativity between boys and girls, as well as to study the changes due to creativity training, split-plot ANOVAs were performed for the two creativity tests and, in the case of the PIC, one was performed for each scale. All analyses were performed with SPSS v21.

## 3. Results

### 3.1. Creative Imagination Test for Children (CREA)

The data for the CREA test were collected from 29 males and 13 females from the first to the third grades of primary school before and after the creativity training. Descriptive statistics by gender (mean and standard deviation) are presented in Table 3.

The assumption of equality of covariances was not met (Box’M (3.11, 117.175 = 9.089; F = 2.823; *p* = 0.037); however, equality of variances was met (Pre: Levene Test (1.40) = 3.98; *p* = 0.532; Post: Levene Test (1.40) = 1.882; *p* = 0.178). In Table 4 are shown the values of the contrasts. Significant improvements due to creativity training were observed, with a large effect size. Neither the interaction nor the difference between genders was significant.

### 3.2. Creative Imagination Test for Children (PIC_N)

#### Narrative Creativity

Students from fourth to sixth grade were evaluated for their creativity with the PIC-N test. Data were collected from 21 females and 37 males. The results of the descriptive statistics (mean and standard deviation) of the three narrative creativity scales appear in Table 5.

In the split-plot contrast for the Fluency variable, the assumptions of equality of covariances (Box’M (3.52, 262.153 = 0.487; F = 0.926; *p* = 0.881) and equality of variances (Pre: Levene Test (1.56) = 0.220; *p* = 0.641; Post: Levene Test (1.56) = 0.120; *p* = 0.731) were met. The effects are indicated in Table 6. There were significant improvements due to training, with a large effect size, but there were no differences by gender, although the male effect size is small. Interaction was not significant either.

In the comparison performed to study the differences in Flexibility, the assumptions of equality of covariances (Box’M (3.52, 262.153 = 0.695; F = 0.222; *p* = 0.881) and variances (Pre: Levene Test (1.56) = 0.022; *p* = 0.882; Post: Levene Test (1.56) = 0.322; *p* = 0.573) were fulfilled. The effects found in the split-plot comparison are shown in Table 7. There were significant differences in flexibility, with a large effect size, due to the training received, but not in gender (although the male effect size found was small) or in their interaction.

With respect to the Originality variable, the split-plot comparison performed met the assumptions of equality of covariances (Box’M (3.52, 262.153 = 0.684; F = 0.218; *p* = 0.884) and variances (Pre: Levene Test (1.56) = 0.620; *p* = 0.434; Post: Levene Test (1.56) = 0.209; *p* = 0.650). The results of the contrasts are presented in Table 8. There were pre–post differences, pointing to score improvements due to training, with a large effect size. There were no differences due to gender.

Nevertheless, the interaction was significant, with a medium effect size. There were improvements in both groups after training; however, the girls’ results were greater (see Figure 1).

### 3.3. Graphic Creativity

The descriptive statistics (mean and standard deviation) found for the four graphic creativity variables are presented by gender in Table 9.

The split-plot comparison of the Elaboration variable showed that the ANOVA assumptions of equality of covariances (Box’M (3.52, 262.153 = 3.285; F = 1.047; *p* = 0.370) and variances (Pre: Levene Test (1.56) = 2.923; *p* = 0.093; Post: Levene Test (1.56) = 2.579; *p* = 0.114) were fulfilled. In Table 10 can be observed the results of the comparisons. There were pre–post differences, pointing to score improvements due to training, with a large effect size. There were no differences due to gender, nor was the interaction significant, but in both cases, the effect sizes were small and medium, respectively.

For the Shadows and Color variable, there were no violations of the ANOVA assumptions of equality of covariances (Box’M (3, 52,262.153 = 7.370; F = 2.353; *p* = 0.070) and variances (Pre: Levene Test (1.56) = 1.969; *p* = 0.166; Post: Levene Test (1.56) = 2.407; *p* = 0.126). The results of the split-plot comparison performed are displayed in Table 11. In this case, there were remarkably lower scores after training, with no significant differences in gender or in the interaction, although the effect size was small in the former and medium in the latter.

The split-plot comparison of the Title variable showed that the ANOVA assumptions of equality of covariances (Box’M (3, 52,262.153 = 2.078; F = 0.663; *p* = 0.575) and variances (Pre: Levene Test (1.56) = 0.142; *p* = 0.708; Post: Levene Test (1.56) = 046; *p* = 0.831) were fulfilled. No significant differences were observed in any of the three comparisons (see Table 12).

The split-plot comparison of the Details variable showed that the ANOVA assumptions of equality of covariances (Box’M (3, 52,262.153 = 44.196; F = 14.094; *p* = 0.001) and variances (Pre: Levene Test (1.56) = 15.301; *p* = 0.001; Post: Levene Test (1.56) = 10.021; *p* = 0.003) were not met, so the Pilloi contrast was taken into account because it was the most robust. Significant differences were only observed in gender, with a significantly higher score in the males, presenting a medium effect size. These results are shown in Table 13.

## 4. Discussion

The results presented in this study highlight two issues. First, the performance in creativity was equal among both males and females, since a significant difference was found in only one of the variables (Details), in favor of the males. However, this finding should be taken with caution because, on the one hand, this was the only variable where differences between genders appeared and because the reliability of the variable was very low in the first pass of the test. This leads to the suggestion of analyzing this data in a larger sample to find out whether the low reliability was due to the test itself or to the characteristics of the sample.

In any case, the results of this study are in line with those of a wide range of studies in which no differences between genders were observed in cognitive variables [50,51,52]. This aspect is especially relevant as studies on gender differences have been generalized with all of the relevance that they have at a practical level. For example, the lower presence of women in technical and engineering careers cannot be based on cognitive differences, but is rather due to contextual stereotypes that must be overcome to achieve equity between women and men [43,44,45]. This will allow girls to be able to choose the career of their choice according to their abilities and desires and not on misconceptions that marginalize them [57,58].

The second aspect that emerges from this study is the importance of creativity training. In this way, it becomes clear that this construct is not a characteristic that one either has or does not, but a quality that can be developed if it is properly trained. Creativity is at the basis of any social development, so it is essential that students reach their full potential through training. Moreover, the sample that we dealt with was made up of high-capacity students. The development of creativity thus acquires a special relevance, as it will allow them to obtain relevant and important results for the society in which they live. A particularly relevant fact is that, because ability matching allows efficient progress in the school curriculum at each educational level, it is possible to carry out intervention programs within the school day, without waiting to conduct them outside of school, as with other educational models. This is one of the clear advantages in the education of the most capable by grouping.

Finally, it should be noted that creativity is multimodal [31]. Although important achievements are observed in narrative creativity, the same does not happen with graphic creativity, and this has also been observed in other studies that have used the same test [40,59]. It is widely known in the literature that the label of creativity encompasses a wide variety of concepts. It is essential to bound the different conceptions of creativity and analyze their contributions to different applications, especially in the education field.

## 5. Conclusions

The present study showed that training through an Integral Innovation and Creativity Program contributed to significant improvements in the various components, such as fluency, flexibility, and originality in the narrative area, as well as elaboration, shadows and color, and special details in the graphic area. However, no differences were found according sex, except for the component “special details”, with males scoring higher, so a greater capacity for insight or perceptual restructuring can be attributed to them, that is, a capacity to see a problem differently from how others see it and to establish unions of two or more drawings in a single figure, rotations, inversions, expansions, or some other details that are very striking through their graphic productions.

Taking these results as a reference, it cannot be assured that males are more creative than women; the fact that they scored higher in only one variable highlights the importance of continuing studies with large and diverse samples, which will allow us to establish conclusions with greater scientific rigor and to justify why the components of creativity can be associated more with one gender or the other.

## Figures and Tables

**Figure 1 children-09-01081-f001:**
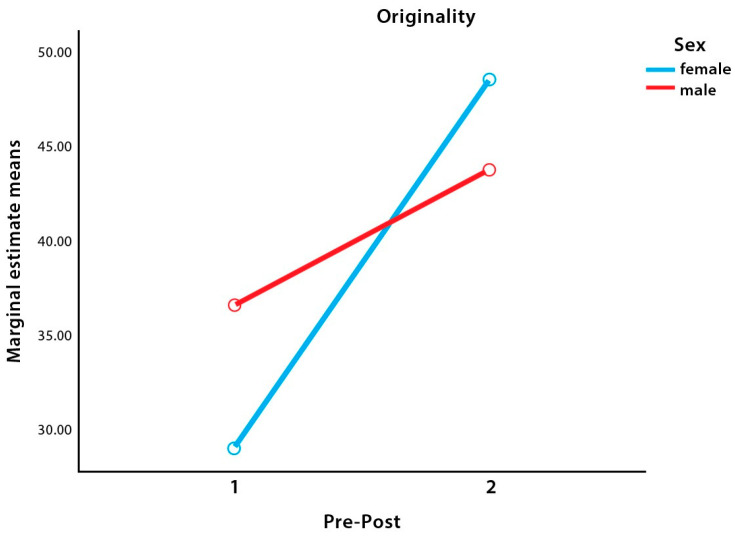
Interaction in Originality.

**Table 1 children-09-01081-t001:** School grade and gender of students who entered CEPAC in the 2017–2018 school year.

	Participants	
School Grade	Women	Men
1	11	19
2	3	12
3	8	7
4	6	9
5	5	10
6	2	13
Total	35	70

**Table 2 children-09-01081-t002:** Reliability of the Creative Imagination Test for Children.

Test	Pretest	Post-Test
CREA	0.806	0.827
PIJ-N	0.790	0.827
Fluency	0.680	0.741
Originality	0.422	0.424
Elaboration	0.715	0.750
Shadows	0.886	0.880
Title	0.786	0.856
Special Details	0.278	0.770

**Table 3 children-09-01081-t003:** Descriptive statistics for CREA (29 men, 13 women).

	Pretest		Post-Test	
	Mean	SD	Mean	SD
Female	3.08	1.55	8.23	4.55
Male	2.86	1.75	8.45	3.05

**Table 4 children-09-01081-t004:** Effects—CREA.

Effects	Mean Square	F (1.40)	*p*	Partial η^2^	Power
Gender	3.158 × 10^−5^	0.000	0.999	0.000	0.050
Error	9.587				
Pre–post	517.696	86.152	0.000	0.683	1.000
Interaction	0.839	0.140	0.711	0.003	0.065
Error	6.009				

**Table 5 children-09-01081-t005:** Descriptive statistics of the PIC-N narrative creativity scales.

**Fluency**				
	**Pretest**		**Post-test**	
	Mean	SD	Mean	SD
Female	40.67	23.43	58.29	27.90
Male	47.81	21.03	61.59	27.49
**Flexibility**				
	**Pretest**		**Post-test**	
	Mean	SD	Mean	SD
Female	19.71	7.61014	27.33	8.97
Male	23.16	7.63	29.00	10.28
**Originality**				
	**Pretest**		**Post-test**	
	Mean	SD	Mean	SD
Female	29.05	17.65	48.52	23.37
Male	36.59	18.53	43.76	20.37

**Table 6 children-09-01081-t006:** Effects on Fluency.

Effects	Mean Square	F (1.56)	*p*	Partial η^2^	Power
Gender	175.212	1.539	0.220	0.027	0.230
Error	113.839				
Pre–post	1212.976	29.572	0.000	0.346	1.000
Interaction	21.252	0.518	0.475	0.009	0.109
Error	41.018				

**Table 7 children-09-01081-t007:** Effects on Flexibility.

Effects	Mean Square	F (1.56)	*p*	Partial η^2^	Power
Gender	175.217	1.539	0.3220	0.027	0.230
Error	113.839				
Pre–post	1212.976	29.572	0.000	0.346	1.000
Interaction	21.252	0.518	0.475	0.009	0.109
Error	41.018				

**Table 8 children-09-01081-t008:** Effects on Originality.

Effects	Mean Square	F (1.56)	*p*	Partial η^2^	Power
Gender	51.764	0.90	0.766	0.002	0.060
Error	577.675				
Pre–post	4753.109	21.987	0.000	0.282	0.996
Interaction	1015.695	4.698	0.034	0.077	0.568
Error	216.181				

**Table 9 children-09-01081-t009:** Descriptive statistics of the graphic creativity variables of the PIC-N.

**Elaboration**				
	**Pretest**		**Post-test**	
	Mean	SD	Mean	SD
Female	2.29	2.15	3.86	1.74
Male	2.30	1.71	3.00	2.17
**Shadows and Color**				
	**Pretest**		**Post-test**	
	Mean	SD	Mean	SD
Female	3.33	2.50	0.67	1.20
Male	2.24	1.79	0.87	1.72
**Title**				
	**Pretest**		**Post-test**	
	Mean	SD	Mean	SD
Female	2.19	1.81	2.62	1.94
Male	2.57	2.07553	2.76	2.22
**Special Details**				
	**Pretest**		**Post-test**	
	Mean	SD	Mean	SD
Female	0.05	0.22	0.10	0.30
Male	0.27	0.56	0.46	1.01

**Table 10 children-09-01081-t010:** Effects on Elaboration.

Effects	Mean Square	F (1.56)	*p*	Partial η^2^	Power
Gender	4.789	0.951	0.334	0.017	0.160
Error	5.038				
Pre–post	34.641	13.247	0.001	0.191	0.947
Interaction	5.055	1.933	0.070	0.033	0.277
Error	2.610				

**Table 11 children-09-01081-t011:** Effects on Shadows and Color.

Effects	Mean Square	F (1.56)	*p*	Partial η^2^	Power
Gender	5.328	1.457	0.232	0.025	0.220
Error	3.657				
Pre–post	109.600	36.171	0.000	0.392	1.000
Interaction	11.117	3.699	0.061	0.061	0.469
Error	3.030				

**Table 12 children-09-01081-t012:** Effects on Title.

Effects	Mean Square	F (1.56)	*p*	Partial η^2^	Power
Gender	1.775	0.301	0.586	0.005	0.084
Error	5.905				
Pre–post	2.556	1.012	0.319	0.018	0.167
Interaction	0.384	0.152	0.698	0.003	0.067
Error	2.525				

**Table 13 children-09-01081-t013:** Effects on Details.

Effects	Mean Square	F (1.56)	*p*	Partial η^2^	Power
Gender	2.307	4.797	0.033	0.079	0.576
Error	0.481				
Pre–post	0.376	0.865	0.356	0.015	0.150
Interaction	0.134	0.309	0.580	0.005	0.085
Error	0.434				

## Data Availability

The data presented in this study are available upon request from the corresponding author, due to restrictions, privacy, and ethical constraints.

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
