# Peer review of "Differences between Creativity and Gender in Students with High Abilities Attending a School with Total Grouping"

_children, 2022, doi:10.3390/children9071081_

Round 1
Reviewer 1 Report
1. The title of this article doesn't well describe the most important research variables, such as student gender, possibility of improving creativity; accordingly, relationships between these variables are unclear, either.
2. More literature review is required to explain why gender is crucial in this study. As the study found the only gender difference of creativity performance is in the variable, Details, the discussion section should explore the reason based on further literature review.
3. The difference between creativity and creative thinking needs to specify in detail. To capture the opportunity of improving creativity, this study should not ignore the effect of in-school training on creative thinking, rather than creativity.
4. The role of in-school training, as a “rough” variable, is not defined or deeply discussed. If in-school training is a treatment in an experiment, we should illustrate the whole relationship between these variables before we compare the pre-test and post-test. For example, there could be some key mediator, moderator, or covariate variable in the relationship.
5. To sum up, the first purpose of this study requires more literature review to present the significance of gender. The second purpose need a deeper discussion to show what possibility of improving creativity among high ability students.
Author Response
Dear reviewer,
Thank you very much for your suggestions and contributions that will help us to enrich the manuscript.
- The title of this article doesn't well describe the most important research variables, such as student gender, possibility of improving creativity; accordingly, relationships between these variables are unclear, either.
The title of the manuscript has been changed: Differences between creativity and gender in students with high abilities attending a school of total grouping.
- More literature review is required to explain why gender is crucial in this study. As the study found the only gender difference of creativity performance is in the variable, Details, the discussion section should explore the reason based on further literature review.
Further studies regarding gender differences have been included in the lines 60-62, 65-68, 92-95 and 102-105.
Discussion should the literature review 292-307.
- The difference between creativity and creative thinking needs to specify in detail. To capture the opportunity of improving creativity, this study should not ignore the effect of in-school training on creative thinking, rather than creativity.
It have been included in the lines 60-62, 65-67, 73-79.
- The role of in-school training, as a “rough” variable, is not defined or deeply discussed. If in-school training is a treatment in an experiment, we should illustrate the whole relationship between these variables before we compare the pre-test and post-test. For example, there could be some key mediator, moderator, or covariate variable in the relationship.
It have been included in the lines 65-68.
In the procedure addresses the importance of creativity training in this study (lines 165-167).
The discussion addresses the importance of creativity training and what it entails (lines 292-300).
- To sum up, the first purpose of this study requires more literature review to present the significance of gender. The second purpose need a deeper discussion to show what possibility of improving creativity among high ability students.
We hope we have been able to respond to your important suggestions.

Reviewer 2 Report
The manuscript deals with in-school training for creativity by measuring several test methods. The manuscript is well-organized and -structured, but there are several issues. Reorganization is required. In other words, some paragraphs are too long and some are too short. Introduction is well-written. Method (manuscript's core concept) is too short and has no enough contribution. More detailed description is required. Throughout the manuscript, there are numerous typos and errors. E.g., captions of figures and tables must be re-written, and some of them have no captions. No conclusion section in the manuscript. References are somewhat old. More recent references are required. With recent references, comparative analysis with the state of the art studies is required. I recommend summarized contributions and a figure that shows the core concept of the research motivation and background in Introduction. Numeric summaries can be added to Abstract and Conclusion. For the result section, the authors' insights and implication can be added, not just describing the result itself. On the whole, the manuscript is incomplete and requires major revisions.
Author Response
Dear reviewer,
Thank you very much for your suggestions and contributions that will help us to enrich the manuscript.
The manuscript deals with in-school training for creativity by measuring several test methods. The manuscript is well-organized and -structured, but there are several issues. Reorganization is required. In other words, some paragraphs are too long and some are too short.
This has been worked on so that they would be balanced.
Introduction is well-written. Method (manuscript's core concept) is too short and has no enough contribution. More detailed description is required.
Some explanations have been included in the method.
Throughout the manuscript, there are numerous typos and errors. E.g., captions of figures and tables must be re-written, and some of them have no captions.
The errors have been corrected.
No conclusion section in the manuscript.
The conclusions have been included
References are somewhat old. More recent references are required. With recent references, comparative analysis with the state of the art studies is required. I recommend summarized contributions and a figure that shows the core concept of the research motivation and background in Introduction.
The references have been included
Further studies regarding gender differences have been included in the lines 60-62, 65-68, 92-95 and 102-105.
Discussion should the literature review 292-307.
Numeric summaries can be added to Abstract and Conclusion.
The following have been included
For the result section, the authors' insights and implication can be added, not just describing the result itself. On the whole, the manuscript is incomplete and requires major revisions.

Round 2
Reviewer 1 Report
Authors have made best to revise, including adding description to the research objectives and contribution. However, the discussion section should be improved by deeper argument against why creativity training is needed according to the result of this study. Since gender does not play role in the participants’ performance of creativity tests, what training programs are needed? What insight comes up with the results? Without such discussion, the paper would be a report of those questionnaires.
Reviewer 2 Report
The authors revised the manuscript based on the previous review.
Thus, I recommend the manuscript for publication.